# Setting Patient-Centered Priorities for Cardiovascular Disease in Central Appalachia: Engaging Stakeholder Experts to Develop a Research Agenda

**DOI:** 10.3390/ijerph20095660

**Published:** 2023-04-27

**Authors:** Dumisa Nyarambi, Fenose Osedeme, Hadii M. Mamudu, Mary A. Littleton, Amy M. Poole, Cynthia Blair, Carl Voigt, Rob Gregory, David Drozek, David W. Stewart, Florence M. Weierbach, Timir K. Paul, Emily K. Flores, Holly Wei

**Affiliations:** 1Department of Community and Behavioral Health, College of Public Health, East Tennessee State University, Johnson City, TN 37614, USA; nyarambid@etsu.edu (D.N.); osedeme@etsu.edu (F.O.);; 2Department of Health Services Management and Policy, College of Public Health, East Tennessee State University, Johnson City, TN 37614, USA; 3Center for Cardiovascular Risk Research, East Tennessee State University, Johnson City, TN 37614, USA; poolea@etsu.edu (A.M.P.); cjblair00769@gmail.com (C.B.); voigt7@comcast.net (C.V.); stewardw@etsu.edu (D.W.S.); weierbach@etsu.edu (F.M.W.); tpaul5@uthsc.edu (T.K.P.); florese@etsu.edu (E.K.F.); 4Karing Hearts Cardiology, Johnson City, TN 37604, USA; rob@karingheartscardiology.com; 5Heritage College of Osteopathic Medicine, Ohio University, Athens, OH 45701, USA; drozek@ohio.edu; 6Department of Pharmacy Practice, Bill Gatton College of Pharmacy, East Tennessee State University, Johnson City, TN 37614, USA; 7College of Nursing, East Tennessee State University, Johnson City, TN 37614, USA; weihl1@etsu.edu; 8College of Medicine, Health Science Center, University of Tennessee-Nashville, Nashville, TN 37203, USA

**Keywords:** patient-centered care, cardiovascular disease, Central Appalachian Region

## Abstract

The disproportionate burden of cardiovascular diseases (CVD) and associated risk factors continues to exist in the Central Appalachian Region (CAR) of the United States. Previous studies to gather data about patient-centered care for CVD in the region were conducted through focus group discussions. There have not been any studies that used a collaborative framework where patients, providers, and community stakeholders were engaged as panelists. The objective of this study was to identify patient-centered research priorities for CVD in the CAR. We used a modified Delphi approach to administer questionnaires to forty-two stakeholder experts in six states representing the CAR between the fall of 2018 and the summer of 2019. Their responses were analyzed for rankings and derived priorities by research gaps. Six of the fifteen research priorities identified were patient-centered. These patient-centered priorities included shorter wait times for appointments; educating patients at their level; empowering patients to take responsibility for their health; access to quality providers; heart disease specialists for rural areas; and lifestyle changes. The participants’ commitments to identify patient-centered research priorities indicate the potential to engage in community-based collaboration to address the burden of CVD in the CAR.

## 1. Introduction

It has been established that disparities in cardiovascular diseases (CVD) and risk factors exist across demographic and socioeconomic groups and geographic locations in the United States (U.S.) [1,2]. A key goal of Healthy People 2030 is to eliminate these disparities and achieve health equity, well-being, and health literacy [3]. Appalachian residents particularly suffer from the high burden of both diagnosed and undiagnosed CVD when compared to residents in non-Appalachian counties [4]. Additionally, studies have indicated that the burden of CVD in the Central Appalachian Region (CAR) is disproportionately greater than in non-CAR regions in the same state and national rates [4,5]. Research indicates that priorities need to be set for healthcare improvement and to develop targeted interventions [6] for patient-centered care. However, research involving CVD and risk factors in the CAR is sparse, providing rationales for this study.

Due to adverse historical experiences and sociocultural characteristics of the CAR [7,8], identifying priorities for CVD research requires engaging diverse community stakeholders [2]. Central Appalachia is a region associated with multiple challenges that are collectively linked to social determinants of health. Some of these problems include lower per capita incomes, higher poverty rates, lower educational attainment, and healthcare-related issues, such as reduced medical care access and a higher prevalence of chronic diseases [9]. The declining coal industry has further negatively impacted the region with problems in social and environmental conditions [9]. These regional problems result in health disparities associated with barriers to CVD management and optimal quality of life for patients with CVD in Appalachian communities [9,10,11]. 

The complexity of the aforementioned health disparities, both in the community and clinical settings, necessitates stakeholder engagement early on in developing research priorities, the implementation process, and later in the dissemination process to ensure the improved translation of research into practice [12]. As guided by PCORI’s conceptualization, our concept of stakeholders refers to providers, patients, clinicians, researchers, purchasers, payers, industry, policymakers, training institutions, hospitals, and health systems [13]. These individuals have a stake and direct interest in the health outcomes of the community being addressed [14]. Previous research, which utilized community-oriented approaches such as principles of community-based participatory research (CBPR), has helped shift the role of the patient in the research process from the subject of research to a stakeholder in research [12,14]. 

Community-based participatory research (CBPR) is an approach to research that utilizes community organizing principles and integrates community perspectives within every research phase [15]. The approach is used in studies aimed at setting research priorities in other regions of the U.S. It is helpful in linking research and practice, focusing on context for creating change [16]. 

The nine principles of CBPR include: (1) recognizing the community as a unit of identity; (2) building on strengths and resources within the community; (3) facilitating the collaborative, equitable involvement of all partners in all phases of the research; (4) integrating knowledge and action for the mutual benefit of all partners; (5) promoting a co-learning and empowering process that attends to social inequalities; (6) involving a cyclical and iterative process; (7) addressing health from both positive and ecological perspectives; (8) disseminating findings and knowledge gained to all partners; and (9) involving a long-term commitment by all partners [5].

Stakeholders have been involved in research that seeks to set priorities for healthcare improvement in low-resource and rural environments. A systematic review of priority setting studies conducted in the U.S. found that most stakeholders engaged in health research priority setting were patients, caregivers, and healthcare providers [17]. This type of collaboration highlights the much-needed place of patients as stakeholder experts and research partners having the opportunity to dialogue with other stakeholders [18,19,20]. 

The value of patient involvement in priority setting is clear because patients have the right to be involved in their healthcare. The equitable involvement of patients in research was recommended in a study conducted on priority setting in kidney disease to ensure that the relevant priorities identified in collaboration with patients were funded [21]. In a pilot study that engaged patients and community stakeholders, the researchers validated moving from an academic researcher-centric approach to one that embraced the knowledge and input of patients [21,22,23]. A study conducted in the U.S. for chronic childhood obesity to describe the priorities of stakeholders including patients, caregivers, and health professionals indicated that 24% of stakeholders were parents and caregivers, and 5% of them were children [24]. Thus, setting priorities for research involves collaboration across the stakeholder spectrum for health issues.

Stakeholders such as patients have played an active role as partners in developing research agendas [19]. Although studies have successfully gathered data on stakeholder preferences through consultative methods such as focus groups, this method lacks a collaborative framework for systematically engaging stakeholders for the purpose of capacity building. No studies have been conducted to set priorities for patient-centered CVD research in the CAR. Additionally, the patient experience of mistrust and skepticism of health professionals, and physician avoidance, have been identified in Appalachian communities [25]. These serve as the impetus for this study.

The purpose of this study was to engage patients, providers, and caregivers with knowledge of CVD in the CAR to identify patient-centered care research priorities for the region. With our research we seek to answer the following research questions: 

What are the patient-centered care research priorities peculiar to the CAR? 

What research priorities can we identify using the principles of CBPR?

The patient-centered model has been recommended and promoted for adoption in the public health field because evidence shows improved health outcomes when implemented [26]. Stakeholder experts, including informed and involved patients, receptive and responsive health professionals, and a collaborative, supportive healthcare environment fit well within the context of patient navigation and patient-clinician relationships where care is delivered [27]. Thus, this study filled a gap in the literature by providing an evidence-based process of engaging stakeholder experts using a modified Delphi method, in setting patient-centered research priorities for CVD in the CAR. The study is unique because it engaged patients from the affected communities across six states as stakeholder experts, and provided new insight into consensus on managing CVD patient populations [28].

## 2. Materials and Methods

### 2.1. Study Participants

Study participants comprised diverse CVD stakeholders involved in the 2018–2019 Patient-Centered Outcomes Research Institute (PCORI) engagement project. 

Stakeholder experts for this project included patients, family/non-professional caregivers, public health professionals, and medical/healthcare providers chosen based on their expertise, membership in a community organization or patient advocacy group, and ability and willingness to participate in the PCORI priority setting project. These criteria resulted in a group of participants representing broad interests for the region.

All participants were provided with information that included the study purpose and contact details of the principal investigator and the project coordinator. In compliance with our Institutional Review Board (IRB) and Helsinki Declaration [22], study participants were notified that the study was voluntary and were informed of efforts in place to ensure confidentiality and privacy. Verbal consent to participate was obtained from each stakeholder.

### 2.2. Study Setting

The study was conducted in Central Appalachia, comprising 228 contiguous and 2 non-contiguous counties in six states: Kentucky, North Carolina, Ohio, Tennessee, Virginia, and West Virginia (Figure 1).

### 2.3. Study Approach 

This study utilized grounded theory and principles of community-based participatory research (CBPR) as approaches to incorporate the strengths and resources of multiple stakeholders involved in addressing the health issue of CVD in the CAR. Out of the nine principles of CBPR, this study utilized four principles: building on strengths and resources within the community, ensuring a cyclical and iterative process, facilitating collaborative, equitable involvement of all partners in all phases of the research, and promoting a co-learning and empowering process that attends to social inequalities [15,17]. 

### 2.4. Study Design

We selected the Delphi method because it allows panelists to express their diverse perspectives on a subject matter in a structured and anonymous way [23]. It also allows for large geographical representation, which was important for covering all the counties in the CAR. 

Ref. [24] The Delphi method encompasses four principles of CBPR: use of a cyclical and iterative process, building on the strengths and resources within the community, facilitating the collaborative and equitable involvement of all partners in all stages of the research, and promoting a co-learning and empowering process that attends to social inequalities. To establish consensus, we administered surveys to address the following research questions:

What are the patient-centered care research priorities peculiar to the CAR? 

What research priorities can we identify using the principles of CBPR?

Stakeholder experts were invited to participate on the Delphi panel via email. Data were collected through open-ended questions, as well as priority rankings. We aimed to use the qualitative data gathered from stakeholders to effectively achieve consensus in establishing priority research areas in the CAR (Table 1).

### 2.5. Data Collection and Analysis

Data were collected between October 2018 and July 2019 through the electronic administration of questionnaires by e-mail to stakeholders in all six states represented by the CAR (Figure 1). The questionnaires were developed from data derived from the environmental scan, focus groups, and CVD disparity statistics for the Central Appalachian communities. The first step was conducting round 1 in an unstructured format to effectively develop a list of stakeholder priorities. Panelists were presented with an open-ended question and asked to provide their top five priorities for CVD in their community and then rank them; 1 was considered the top priority, and 5 was considered the least. Thematic analysis was used to analyze the collected data. The qualitative data were entered into Nvivo, a qualitative data management software in which the grounded theory approach was used to identify themes that support the priorities and rankings.

The second step was a round of data collection conducted by means of a questionnaire. The questionnaire, which had 15 closed-ended questions, was developed using Google forms, with the content of the questionnaire derived from round one responses. The panelists were asked to rank priority items for their respective communities. A total of 42 respondents participated in round 2 (14 patients/non-licensed caregivers, 15 community stakeholders, and 13 providers). The response rate was 52%. 

CVD factors identified for priority ranking covered factors such as nutrition, physical activity, education, rural community outreach, health care access, quality of care, patient-provider communication, cost of medication, focus on preventive medicine, and tobacco use (Table 1). Panelists were asked to identify themselves as either patients, non-licensed caregivers, community stakeholders, or providers/professionals. The panelists were asked to choose their top three to four priorities from the respective questions for a total of 24 priorities each. During the response period, e-mail reminders were sent to panelists every two weeks. Data were analyzed using descriptive statistics in Microsoft Excel and presented as percentages with illustrations in bar graph format. The top 24 priorities established in round two were concretized and used to develop the questionnaire for round 3. 

The third step was round 3, which was the final round. A paper-based questionnaire was administered at the annual CVD Appalachia Conference in August 2019. The top priorities generated in round 2 were presented to conference stakeholders to prioritize. Each closed-ended question asked for a priority ranking, with only one priority being selected from each question. Of 40 stakeholder experts who agreed to complete the questionnaire, 31 submitted their responses (7 patients, 18 providers, and 6 community stakeholders). The response rate for round 3 was 77.5%. The modified Delphi method allows for questionnaire administration in a forum where participants are physically present [2]. Data were analyzed using mixed methods of thematic analysis (Nvivo) and quantitative analysis with descriptive statistics for the rankings.

## 3. Results and Outcomes

### CVD Patient-Centered Research Priorities

Fifteen priority items for CVD care in the CAR were identified by forty-two stakeholder experts (Table 2). 

The modified Delphi outcome summary for each round, is shown in Table 3. 

Six of the fifteen priority items were patient-centered, and mapped to two specific areas: (1) patient-centered care; and (2) managing CVD in patient populations (Table 4).

The consensus response rate was 77.5%. The most highly ranked priorities were divided into the following categories: overall access to quality healthcare, patient-provider communication, lifestyle modifying techniques, strategies for preventing heart disease, and medication affordability.

## 4. Discussion

To our knowledge, no other studies on setting priorities for a CVD research agenda in the CAR have been found. This gap in the literature about prioritizing research in Appalachia highlights the relevance of this study. Disparities exist in CVD risks and outcomes in the U.S., and several barriers to managing CVD have been associated with living in the Appalachian region [4,27]. Moreover, individuals experiencing the greatest burden of CVD may not accept or understand the need for lifestyle changes and may not have access to medical care. The combination of acceptance and understanding requires a comprehensive approach to patient-centered care originating from stakeholder engagement. [29]. Previous studies discuss various approaches to patient-centered care, such as patient-provider communication focusing on community health worker interventions [9,30,31]. However, no studies documenting stakeholder priority setting for the CVD research agenda were found in the CAR. The results of this study converge with literature that calls for a patient-centered approach to health promotion and the reduction of health inequalities and disparities, especially in rural populations [32]. 

In an environment of mistrust for medical providers, the patient-centered care approach provides an atmosphere to build trust between patients and providers [19]. Patient reflections on patient-centered care indicate that trust remains an important part of patient-provider relationships [19,29,30]. While trust may be defined in numerous ways, there is consensus in the literature that it entails believing that providers have the patient’s best interest and will act in goodwill [31]. The trust deficit results in barriers to relationships with providers and impending patient adherence to medical recommendations and their utilization of preventive health services [25,33,34]. A study in Appalachia found that distrust in the healthcare system influences patients against taking preventive health measures [35].

The results of this study illustrate the broad range of opportunities for patient-centered, evidence-based interventions. The unique contribution of our study stems from addressing the gap in the literature. Our results confirm and enhance the patient-centered care and CVD literature by demonstrating that setting priorities for patient-centered care within disease-specific populations across large geographic areas is desired and feasible [17,31,35].

This study also (1) confirms that engaging patients and community members in rural and underserved areas as stakeholders for consensus in the research process is achievable; and (2) provides a deeper understanding of patient perspectives of how CVD populations are managed in local communities [36]. This study implies an increasing expectation from stakeholders to recognize the “patient as a person” rather than seeing the person as a patient. To see the “patient as a person”, emphasis must be on delivering patient-driven proactive and personalized care, and also understanding the patient and their illness in the context of the individual [37,38]. Emerging literature has recommended a patient-centered care approach for patients with chronic diseases with the context of the whole person taken into consideration [37]. To further exemplify the “patient as a person” in priority setting acknowledges the patient as a key stakeholder in the process, which may impact research outcomes [39]. With this approach, the patient is not only a recipient of treatment and services, but is also an expert engaged in identifying priorities for their care in their community. 

The strengths of this study are the participation of a highly diverse group of stakeholder experts who represented Appalachian counties in six states, the CAR. Utilizing principles of CBPR, patients and family caregivers participated in the study by means of an existing method of priority setting, the modified Delphi method. This strategy supports the engagement of those receiving care to take an active role in their health care by becoming researchers and not merely recipients of care. Stakeholder experts were found to be highly motivated to participate in identifying health disparities in the CAR. The consensus reached after round 3 indicates that providers should come alongside patients to ensure that their needs are met, and contribute to the planning of the disease management process. 

### Limitations

One of the study’s limitations is that the panelists did not receive compensation for their time participating in the study. Logistical limitations also included a limited timeframe to complete the Delphi process. Both these factors may explain the low response rate in round 1. Nonetheless, the themes across these three data collection rounds were consistent.

## 5. Conclusions

This study aimed to fill a gap in priority setting for a CVD research agenda in the CAR. Our results suggest that the modified Delphi method successfully engaged stakeholders as experts in identifying research priorities for the region, which is rural and under-resourced. The priorities included the patient’s voice and perspective, which were identified through an iterative process based on CBPR principles. Six patient-centered priorities included research that supports: (1) access to quality healthcare providers; (2) communicating and providing education to patients on their level; (5) shorter wait times for scheduling appointments; and (6) empowering and motivating patients to take responsibility for their health. Thus, this study provides a platform for future studies to facilitate patient-centered care and patient-centered outcomes research.

## Figures and Tables

**Figure 1 ijerph-20-05660-f001:**
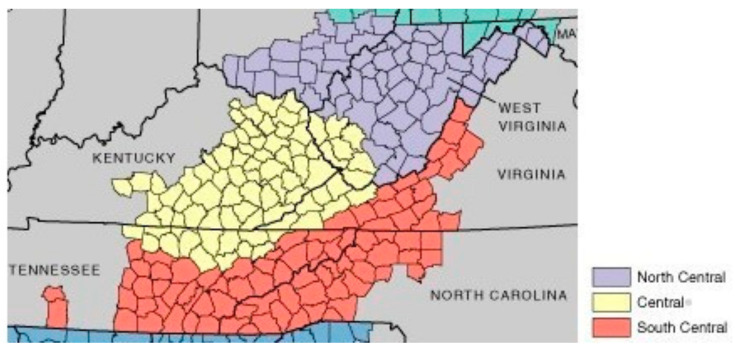
Priority setting study area—Central Appalachia (Source: Appalachian Regional Commission; ARC).

**Table 1 ijerph-20-05660-t001:** Modified Delphi Methodology Summary.

Round Number and Activity/Goal	Methodology/Process
Round 1. Stakeholder experts vote for top five priorities	Participants were asked to compile and rank their top five priorities for cardiovascular disease, based on their expertise or experience. Thematic analysis was used to analyze the collected data. The response rate was 14%.
Round 2. Ranking of Priorities found within qualitative data	Using a 15-question instrument, experts were asked to select a specific number of their top research priorities based on the following categories: (1) Patient-centered care; (2) CVD risk factors; (3) Psychosocial determinants of CVD; (4) Social determinants of CVD. A total of 42/80 experts responded to Round II questionnaire (14 patients/non-licensed caregivers, 15 community stakeholders, and 13 professional/providers) The response rate was 52%.
Further narrowing of priorities	Round II data analyzed. Data were analyzed using descriptive statistics in MS Excel and presented as percentages with illustrations in bar graph format
Round 3. CVD Sharing Session at CVD Appalachia Conference II and Delphi Round III. Discussion and validation	Participants gathered at a conference to determine what priorities would move forward to make up the final research agenda. A paper-based survey was administered. Each closed-ended question asked for a priority ranking, with only one priority being selected from each question. Thirty-one out of forty stakeholders (7 patients, 18 providers, and 6 community stakeholders) submitted their responses. The response rate was 77.5%.
Alignment of research agenda with PCORI initiatives	Fifteen priorities mapped to major areas. Six of the 15 priorities were patient-centered

**Table 2 ijerph-20-05660-t002:** Characteristics of Stakeholder Experts for the Modified Delphi Panel.

Characteristics	Round 1 *n* (%)	Round 2 *n* (%)	Round 3 *n* (%)
State			
Kentucky	0 (0)	3 (7)	0 (0)
North Carolina	0 (0)	2 (5)	2 (6)
Ohio	1 (10)	5 (12)	3 (10)
Tennessee	6 (54)	22 (52)	15 (48)
Virginia	1 (10)	7 (17)	6 (19)
West Virginia	3 (27)	3 (7)	5 (17)
Stakeholder Type			
Patient/Non-licensed caregiver	7 (64)	14 (33)	7 (23)
Provider/Professional	2 (18)	13 (31)	18 (58)
Community stakeholder	2 (18)	15 (36)	6 (19)
Demographic			
Sex			
Male	2 (18)	13 (31)	14 (45)
Female	9 (82)	29 (69)	17 (72)

**Table 3 ijerph-20-05660-t003:** Modified Delphi Outcome Consensus.

Round Number and Goal	Outcomes
Round 1. Stakeholder experts vote for top five priorities	Fifty-five priorities were obtained from 11 experts. The priorities were then grouped into 7 major focus areas. The top five priorities generated: (1) nutrition education and best practices; (2) education for heart disease prevention; (3) increased opportunities for free and no-cost exercise and physical activities; (4) reducing cost of healthcare for heart disease; and (5) treatment of heart disease and stroke
Round 2. Ranking of priorities found within qualitative data	A parsimonious set of priorities was selected by 42 experts from each category. Data ready for analysis to determine the research agenda twenty-four (24) items identified.
Round 3. CVD Sharing Session at CVD Appalachia Conference II and Delphi Round III. Discussion and validation	Further input from 31 experts to determine final priorities. Refined list derived. Fifteen (15) items were identified; six of the 15 priorities were patient-centered as shown in Table 2
Alignment of research agenda with PCORI initiatives	Research agenda established by a consensus of Central Appalachia community stakeholders

**Table 4 ijerph-20-05660-t004:** Priorities Identified at Each Round.

Priority Areas	Round 1	Round 2	Round 3
(1) Patient-centered care	Education Lifestyle	More time with healthcare providers Better communication between providers and patients Shorter wait times for scheduling appointments Communicate and provide education to patients on their level	Communicate and provide education to patients on their level Shorter wait times for scheduling appointments Empower and motivate patients to take responsibility for their health and illnesses
(2) CVD risk factors	Nutrition	Prevention of heart disease education Knowledge and education about the prevention of heart disease and risk factors Knowledge and skills in preparation of heart-healthy foods Knowledge and skills in reducing intake of sodium Knowledge and skills to prevent tobacco use Regular screenings for blood pressure, glucose and cardiac calcium	Knowledge and education about the prevention of heart disease and risk factors Prevention of heart disease education and skills on prevention of risk factors
(3) Psychosocial determinants of health	Prevention	Knowledge and skills on how to reduce and manage stress Skills to prevent nicotine addiction	Knowledge and skills on how to reduce and manage stress Skills to prevent nicotine addiction
(4) Social determinants of health	Physical Activity Nutrition	Access to affordable healthy food Access to care for heart disease Increase availability of no-cost walking options and exercise places in the community Lower costs for uninsured	Knowledge and education about physical activities for all ages
(5) CVD policies and programs	Lifestyle Education Cost	Programs that target high-risk and low-socioeconomic populations to promote healthy lifestyles Affordable public transportation to access healthy lifestyle programs and sites	Programs that target high-risk and low-socioeconomic populations to promote healthy lifestyles School budgets to promote physical education, nutrition, and health education
(6) Medication adherence	Cost Education	Better information and explanation of medications Affordable medications	Affordable medications
(7) Managing CVD patient population	Education Lifestyle Prevention Screening	Healthcare providers to conduct community outreach and/or outreach practices Availability of mobile units to screen and treat heart disease Heart disease specialists in rural areas Empower/motivate patients to take responsibility for illness/health	Access to quality healthcare providers Heart disease specialists in rural areas More information on lifestyle changes versus relying solely on medications

## Data Availability

Not applicable.

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
