# Peer review of "Setting Patient-Centered Priorities for Cardiovascular Disease in Central Appalachia: Engaging Stakeholder Experts to Develop a Research Agenda"

_ijerph, 2023, doi:10.3390/ijerph20095660_

Round 1

Reviewer 1 Report (Previous Reviewer 2)

I would like to commend the authors on the thoughtful consideration of the comments from the previous reviews.

The research questions are now clearly stated. In addition, CBPR and the Delphi processes are much more clearly integrated given the desire for outcome consensus. I would encourage you to consider the language around Results and whether or not it should be Results & Outcomes and that Table 3 be Modified Delphi Outcomes or Outcome Consensus because it is more than an Output Summary given that they resulted from outcome consensus.

As a general practice, references are not usually included in the Conclusion which is usually 2-3 sentences but I think that this works well. 

Spelling/context consideration in 2.3 Study Approach - did you mean CBPR as rather than a approaches...?

Congratulations on engaging stakeholder experts to co-create a meaningful research agenda with you.  

Author Response

Cover Letter

In order to address the gaps identified by the reviewer, suggested  revisions have been made to the manuscript. Attached you will find our revised manuscript and responses to the reviewer’s comments.

  • The manuscript has been revised to have Results and Outcomes section
  • The language around results has been revised to be Outcomes rather than Outputs.
  • Table 3 has been revised to be Modified Delphi Outcome Consensus
  • The spelling/context consideration in 2.3 Study Approach has been corrected to read …CBPR as approaches

Comment
I would like to commend the authors on the thoughtful consideration of the comments from the previous reviews.
Response
Thank you for acknowledging our thoughtful consideration in response to the previous reviewers.

Comment
The research questions are now clearly stated. In addition, CBPR and the Delphi processes are much more clearly integrated given the desire for outcome consensus. I would encourage you to consider the language around Results and whether or not it should be Results & Outcomes and that Table 3 be Modified Delphi Outcomes or Outcome Consensus because it is more than an Output Summary given that they resulted from outcome consensus.
Response
The manuscript has been revised to use the title Results and Outcomes. In addition, Table 3 title has been revised to  Modified Delphi Outcomes  as suggested.  The language around results has been revised to be Outcomes rather than Outputs.

Comment
As a general practice, references are not usually included in the Conclusion which is usually 2-3 sentences but I think that this works well. 
Response
We included references in the Discussion section of the manuscript. No references are included in our Conclusion

Comment
Spelling/context consideration in 2.3 Study Approach - did you mean CBPR as rather than a approaches...?
Response
The referenced spelling/context consideration has been corrected to read …CBPR as approaches…

Comment
Congratulations on engaging stakeholder experts to co-create a meaningful research agenda with you.  
Response
Thank you!

This manuscript is a resubmission of an earlier submission. The following is a list of the peer review reports and author responses from that submission.

Round 1

Reviewer 1 Report

The methodology section requires the attention.

Comment
The methodology section requires the attention.
-- It indicated that three methods namely, Delphi, CBPR, and mixed methods were used but it is not clearly indicated how these methods were used to guide the study and data collection.
-- Which approach of the mixed methods were applied to this study?
Response
Thank you for the feedback! Revisions have been made to clearly indicate how the methods were used. The Delphi was the choice of methodology as it can engage experts without geographical restrictions, which also allows a wide community representation. Developed by the RAND Corporation in the 1950s, the Delphi method has been used to solicit the opinion of a group of experts in an attempt to reach a consensus in several areas,1 including health, health behaviors, and healthcare.2–10 We adapted the modified version of the Delphi method, which has been previously used for studies involving stakeholders pertinent to our study,2,4,6,10–12 to guide our study. Indeed, the approach that we utilized in this study has been validated in a prior study,12 hence, consistent with the literature.

Informed by the modified-Delphi method,2,4,6,10,11,13 we collected the opinions of our expert stakeholders, which comprised of patients and non-licensed caregivers, providers (including clinicians and nurses), representatives of primary care practices, specialties, and patient organizations; and community-based leaders, were gathered through a series of surveys  (Round 1 and 2) that collected both qualitative and quantitative data. Microsoft Excel was used to analyze the data and rank the issues. The Round 3 of data collection of the opinions on the expert stakeholders was conducted during the 1st CVDAppalachia Conference, the PCORI-funded conference that brought over 200 CVD stakeholders (patients, providers, caregivers, community-based groups/leaders, hospitals/healthcare systems, etc.) from six Central Appalachian states (Kentucky, Ohio, North Carolina, Tennessee, Virginia, and West Virginia) to discuss the development of CVD research agenda for the region. The modified Delphi method makes provision for both online and in-person data collection;2,4,6,10,11,13 therefore, our data collection approach was in conformity with the methodology.

The modified Delphi method was complemented by some principles of Community-Based Participatory Research (CBPR) approach to get more community perspectives of patient-centered care among our expert stakeholders in the Central Appalachian region. There are nine principles underlying the CBPR approach, which include 1) recognizing the community as a unit of identity, 2) building on strengths and resources within the community, 3) facilitating collaborative, equitable involvement of all partners in all phases of the research, 4) integrating knowledge and action for the mutual benefit of all partners, 5) promoting a co-learning and empowering process that attends to social inequalities, 6) involving a cyclical and iterative process, 7) addressing health from both positive and ecological perspectives, 8) disseminating findings and knowledge gained to all partners, and 9) involving a long-term commitment by all partners.14–16

We utilized four of these principles to support our modified Delphi method: 1) building on strengths and resources within the community, 2) use of a cyclical and iterative process, 3) facilitating collaborative and equitable involvement of all partners in all stages of research and 4) promoting a co-learning and empowering process that attends to social inequalities. We used this approach because it has a focus on “community ownership,” particularly with populations who experience disparities and may have a mistrust for the healthcare system. This is the case in the Central Appalachian region, and therefore, we ensured that the stakeholders we collaborated with were vested in the community needs, norms and culture with lived experiences of cardiovascular disease. We achieved this by engaging stakeholders, including patients, caregivers, providers, researchers and other community stakeholders. The implication and significance of this was the success in engaging a wide and diverse group of stakeholders in the research process.

In sum, the modified Delphi method, along with principles of CBPR, that we utilized for this study has been used for health-related studies in the past and it is supported by the extant literature. We have included some of this information in the methods section of the revised manuscript.

Reviewer 2 Report

This manuscript, “Setting patient-centered priorities for cardiovascular disease in Central Appalachia: Engaging key partners to develop a research agenda”, will be of interest to clinicians and communities that support and provide health services to underserved communities.

There are several gaps in this manuscript which I will highlight below:

1.     I am a bit confused about the language in the Introduction in that both key partners and stakeholders are used. The authors go on to talk about the value of patient involvement in priority setting but they do not seem to be recognized as experts e.g. authors.

2.     I wondered if the statement “CAR is a region associated with negative social and environmental problems, partly because of the declining coal industry.” is appropriate as the authors then go on to talk about the social determinants of health and I wondered if the language could be transformed to reflect that the region is impacted by the social determinants of health because of the declining coal industry which then does not blame but explains. Thoughts?  

3.     The purpose of the study was to engage patients, providers, and caregivers with knowledge of CVD in the in the CAR to identify patient-centered care research priorities for the region. That being said, the research questions were not clearly identified, and I wondered whether the work would be evidence-informed or evidence-based.

4.     It will be important to link the Delphi method with community-based participatory research (CBPR) because they do not seem to be at this point. From my reading of the “Study Design”, there are three separate methods that need to be linked in a better way – the Delphi method, which is usually qualitative, CBPR, mixed-methods with some reference to grounded theory. Table 1. which is about the Modified Delphi Methodology Summary appears in the article before Data Collection and the Analysis but includes Outputs. As a reader this is very confusing for me; thus, I would encourage the authors to engage the patients and their families to find a better way to tell this story. This would then facilitate member-checking of qualitative research rigour.  

5.     In the Results Section, there are both results and findings. Priority Areas in Round 3 specifically state that Patient-Centered Care should empower and motivate patients to take responsibility for their health and illness. This may work if they have a voice and are heard but this is not congruent throughout the document which goes back to a better description of the Methods and how CBPR is integrated into the Modified Delphi Methodology.

6.     There was very little linkage between the Literature Review, Methods and the Discussion, but it would have been an opportunity to weave CBPR throughout the document which then has the potential to empower/motivate individuals/patients to take responsibility for their health and well-being. For this to occur, it does require that the health care providers also transform to include the individuals/patients in the management planning of the disease process.

7.     The Conclusion needs to evolve from the data and the statements seem to be about getting the individuals/patients to take responsibility for their health and well-being. This may be correct, but it appears to be top-down rather than framed within the context of CBPR which the authors indicate was used.  

I think that this work is important but the authors need to consider and address the aforementioned concerns.

Comment

This manuscript, “Setting patient-centered priorities for cardiovascular disease in Central Appalachia: Engaging key partners to develop a research agenda”, will be of interest to clinicians and communities that support and provide health services to underserved communities.

There are several gaps in this manuscript which I will highlight below:

Response

Thank you! We have diligently revised the manuscript to address the gaps highlighted as suggested by the reviewer.

Comment

  1.    I am a bit confused about the language in the Introduction in that both key partners and stakeholders are used. The authors go on to talk about the value of patient involvement in priority setting but they do not seem to be recognized as experts e.g. authors.

Response

Revisions have been made to avoid confusion on using multiple terms( key partners and stakeholders) Similar to previous studies, our focus was on stakeholders.2,11,12,17,18 As such, we have revised the entire manuscript to consistently use the term stakeholders. We have; therefore, updated the title of the revised manuscript to remove “key partners” and replaced it with experts as a reflection of the Delphi methodology engaged for the study. Revisions were made to provide clarity in the introduction and throughout the manuscript. In addition, definitions of terms have been added for clarity. As in prior studies,2,12,13,17–19 the patients are recognized as expert stakeholders in identifying the priorities. Patients were involved not only as part of the expert stakeholders interviewed but also as authorship of the manuscript, which exemplifies CBPR Principle 3 of facilitating collaborative, equitable involvement of all partners in all phases of the research.14

Comment

  1.    I wondered if the statement “CAR is a region associated with negative social and environmental problems, partly because of the declining coal industry.” is appropriate as the authors then go on to talk about the social determinants of health and I wondered if the language could be transformed to reflect that the region is impacted by the social determinants of health because of the declining coal industry which then does not blame but explains. Thoughts? 

Response

The Centers of Disease Control and Prevention (CDC) defines Social Determinants of Health (SDOH) as conditions in which people are born, grow, work, live, and age, and the wider set of forces and systems shaping the conditions of daily life”.20 The work environment is one of the major determinants of health20–22 and the relationship between coal mine and adverse health outcomes has been documented in the extant literature.23,24 Thus, while there may be immediate economic benefits that may accrue to the workers and mining companies, adverse health outcomes to miners such as respiratory diseases25,26 and other negative externalities from coal mining23,27 have been documented. As such, we have revised the sentence as “CAR is a region associated with negative social and environmental problems” and have deleted “partly because of the declining coal industry” as there is not enough evidence to substantiate that.

 Comment

  1.    The purpose of the study was to engage patients, providers, and caregivers with knowledge of CVD in the in the CAR to identify patient-centered care research priorities for the region. That being said, the research questions were not clearly identified, and I wondered whether the work would be evidence-informed or evidence-based.

Response

As indicated in the response to Reviewer #1 above, we utilized the modified Delphi method, along with principles of CBPR approach for this study. This involved soliciting the opinion of stakeholders to identify priorities of patient-centered care in the CAR region. This study was guided by two research questions which are:

  1. What are the patient-centered care research priorities peculiar to the CAR?
  2. What priorities can we identify using principles of CBPR: building on strengths and resources within the community, ensuring a cyclical and iterative process, facilitating collaborative, equitable involvement of all partners in all phases of the research, and promoting a co-learning and empowering process that attends to social inequalities?

We have clearly stated these research questions in the introduction and method sections of the revised manuscript.

Comment

  1.    It will be important to link the Delphi method with community-based participatory research (CBPR) because they do not seem to be at this point. From my reading of the “Study Design”, there are three separate methods that need to be linked in a better way – the Delphi method, which is usually qualitative, CBPR, mixed-methods with some reference to grounded theory. Table 1. which is about the Modified Delphi Methodology Summary appears in the article before Data Collection and the Analysis but includes Outputs. As a reader this is very confusing for me; thus, I would encourage the authors to engage the patients and their families to find a better way to tell this story. This would then facilitate member-checking of qualitative research rigour.  

Response

The main methodology for this study is the modified Delphi method (see Reviewer #1 above), which involved seeking the opinion of a group of stakeholders in an iterative manner to arrive at a consensus. We used a combination of online and in-person data collection approach using survey questionnaire to collect both qualitative and quantitative information on patient-centered care from stakeholders in the CAR region. As indicated earlier, four (4) principles of the CBPR approach (not methodology) were used in this process. Thus, to clarify, we use mixed methods because the Delphi process is primarily qualitative and involves surveys for priority rankings, as well as open-ended questions to solicit the stakeholders’ opinions about patient-centered care. While the quantitative data were entered into Microsoft Excel for analyze and ranking, the qualitative data were entered into NVivo, a qualitative data management software, where the grounded theory approach28–30 was used to identify themes that support the priorities and the rankings. We have included these linkages in the methods section of the revised manuscript.

Regarding the confusion created by Table 1, we agree with the review, and it was our attempt to limit the number of tables in the manuscript. As such, we have rectified this confusion by removing the Outputs from the Methodology/Process table and adding a 4th table in the Results section of the paper that provides an Output Summary on its own as Table 3.

Comment

  1.    In the Results Section, there are both results and findings. Priority Areas in Round 3 specifically state that Patient-Centered Care should empower and motivate patients to take responsibility for their health and illness. This may work if they have a voice and are heard but this is not congruent throughout the document which goes back to a better description of the Methods and how CBPR is integrated into the Modified Delphi Methodology.

Response

This study was funded by the Patient-Center Outcomes Research Institute (PCORI) in the United States, and one of its key requirements for funding projects is the involvement of patients from the conceptualization of the project to the dissemination of the findings. As such, patients have been involved in all aspects of this study. Specifically, patients have been involved as our expert stakeholders, members of the investigation team, and part of the authorship of this manuscript; therefore, patients were strongly engaged throughout the study. PCORI patient engagement strategies include reaching out to those with a lived experience of a specific diseases, as well as the caregivers/family members who provide their care.14

The stakeholders who engaged in the study included patients and represented six counties of the Central Appalachian community.  At each round of the modified Delphi, patients participated in the priority setting, and their participation gave them a voice. Among these stakeholders were also physicians and unlicensed care givers of CVD patients. Each of these three groups were considered expert stakeholders and engaged together to set the priorities that were obtained by the end of the third round. Principles of the CBPR approach including a cyclical and iterative process; engaging community; facilitating collaboration, and equitable involvement of all stakeholder are all part of the modified Delphi process, with  the use of grounded theory approach to identify supporting themes.31

Comment

  1.    There was very little linkage between the Literature Review, Methods and the Discussion, but it would have been an opportunity to weave CBPR throughout the document which then has the potential to empower/motivate individuals/patients to take responsibility for their health and well-being. For this to occur, it does require that the health care providers also transform to include the individuals/patients in the management planning of the disease process.

Response

The manuscript has been revised to illuminate the linkages between the sections. CBPR has been explained and illustrated in the context of the use of a modified Delphi methodology.

Comment

  1.    The Conclusion needs to evolve from the data and the statements seem to be about getting the individuals/patients to take responsibility for their health and well-being. This may be correct, but it appears to be top-down rather than framed within the context of CBPR which the authors indicate was used.

Response

The crux of this study is using the modified Delphi method to identify and rank priorities involved in patient-centered care by stakeholders, including patients and non-licensed caregivers, in the CAR region. Thus, the data we collected included the voice of patients so are the inferences we make in the conclusion. However, due to this concern raised by the reviewer, we have double-checked to ensure that the conclusions are deduced from the collected data. Thank you!.

Comment 

I think that this work is important but the authors need to consider and address the aforementioned concerns.

Response

Thank you for the constructive feedback! We have revised the manuscript as suggested by the reviewers.

References

  1. RAND Corporation. Delphi Method. https://www.rand.org/topics/delphi-method.html?page=1. Accessed March 12, 2023.
  2. Khodyakov D, Park S, Hutcheon JA, Parisi SM, Bodnar LM. The impact of panel composition and topic on stakeholder perspectives: Generating hypotheses from online maternal and child health modified-Delphi panels. Heal Expect  an Int J public Particip Heal  care Heal policy. 2022;25(2):732-743. doi:10.1111/hex.13420
  3. Shearer AL, Bromley E, Bonds C, Draxler C, Khodyakov D. Improving Mental Health Guardianship: From Prevention to Treatment. Psychiatr Serv. 2022;73(6):642-649. doi:10.1176/appi.ps.202100020
  4. King C, Arnold R, Dao E, et al. Consensus-based approach to managing opioids, including opioid misuse and opioid use disorder, in patients with serious illness: protocol for a modified Delphi process. BMJ Open. 2021;11(5):e045402. doi:10.1136/bmjopen-2020-045402
  5. Khodyakov D, Jilani SM, Dellva S, Faherty LJ. Informing the Development of a Standardized Clinical Definition of Neonatal Abstinence Syndrome: Protocol for a Modified-Delphi Expert Panel. JMIR Res Protoc. 2021;10(9):e25387. doi:10.2196/25387
  6. Grant S, Armstrong C, Khodyakov D. Online Modified-Delphi: a Potential Method for Continuous Patient Engagement Across Stages of Clinical Practice Guideline Development. J Gen Intern Med. 2021;36(6):1746-1750. doi:10.1007/s11606-020-06514-6
  7. Campbell M, Moore G, Evans RE, Khodyakov D, Craig P. ADAPT study: adaptation of evidence-informed complex population health interventions for implementation and/or re-evaluation in new contexts: protocol for a Delphi consensus exercise to develop guidance. BMJ Open. 2020;10(7):e038965. doi:10.1136/bmjopen-2020-038965
  8. Khodyakov D, Chen C. Nature and Predictors of Response Changes in Modified-Delphi Panels. Value Heal J Int Soc Pharmacoeconomics  Outcomes Res. 2020;23(12):1630-1638. doi:10.1016/j.jval.2020.08.2093
  9. Khodyakov D, Chen C. Response changes in Delphi processes: why is it important to provide high-quality feedback to Delphi participants? J Clin Epidemiol. 2020;125:160-161. doi:10.1016/j.jclinepi.2020.04.029
  10. Rubenstein L, Hempel S, Danz M, et al. Eight Priorities for Improving Primary Care Access Management in Healthcare Organizations: Results of a Modified Delphi Stakeholder Panel. J Gen Intern Med. 2020;35(2):523-530. doi:10.1007/s11606-019-05541-2
  11. Khodyakov D, Grant S, Denger B, et al. Practical Considerations in Using Online Modified-Delphi Approaches to Engage Patients and Other Stakeholders in Clinical Practice Guideline Development. Patient. 2020;13(1):11-21. doi:10.1007/s40271-019-00389-4
  12. Khodyakov D, Grant S, Meeker D, Booth M, Pacheco-Santivanez N, Kim K. Comparative Analysis of Stakeholder Experience with an Online Approach to Prioritizing Patient-Centered Research Topics. https://www.rand.org/pubs/external_publications/EP67053.html. Published 2017. Accessed March 12, 2023.
  13. Khodyakov D, Grant S, Barber CE, Marshall DA, Esdaile JM, Lacaille D. Acceptability of an Online Modified Delphi Panel Approach for Developing Health Services Performance Measures. Journal of evaluation in clinical practice. https://www.rand.org/pubs/external_publications/EP66653.html. Published 2016. Accessed March 12, 2023.
  14. Kwon SC, Tandon SD, Islam N, Riley L, Trinh-Shevrin C. Applying a community-based participatory research framework to patient and family engagement in the development of patient-centered outcomes research and practice. Transl Behav Med. 2018;8(5):683-691. doi:10.1093/tbm/ibx026
  15. Wallerstein N, Duran B. Community-based participatory research contributions to intervention research: the intersection of science and practice to improve health equity. Am J Public Health. 2010;100 Suppl 1(Suppl 1):S40-6. doi:10.2105/AJPH.2009.184036
  16. Israel BA, Schulz AJ, Parker EA, Becker AB. Review of community-based research: assessing partnership approaches to improve public health. Annu Rev Public Health. 1998;19:173-202. doi:10.1146/annurev.publhealth.19.1.173
  17. Urk F, Grant S, Bonell C. Involving Stakeholders in Programme Theory Specification: Discussion of a Systematic, Consensus-Based Approach. https://www.rand.org/pubs/external_publications/EP67534.html. Published 2016. Accessed March 12, 2023.
  18. Lapane KL, Dubé C, Hume AL, et al. Priority-Setting to Address the Geriatric Pharmacoparadox for Pain Management: A Nursing Home Stakeholder Delphi Study. Drugs Aging. 2021;38(4):327-340. doi:10.1007/s40266-021-00836-8
  19. Khodyakov D, Grant S, Denger B, et al. Practical Considerations in Using Online Modified-Delphi Approaches to Engage Patients and Other Stakeholders in Clinical Practice Guideline Development. doi:10.1007/s40271-019-00389-4
  20. U.S. Centers for Disease Control and Prevention (CDC). Social Determinants of Health at CDC. https://www.cdc.gov/about/sdoh/index.html. Accessed March 12, 2023.
  21. Sattler B. Farmworkers: Environmental Health and Social Determinants. Annu Rev Nurs Res. 2019;38(1):203-222. doi:10.1891/0739-6686.38.203
  22. Cho M, Lee Y-M, Lim SJ, Lee H. Factors Associated with the Health Literacy on Social Determinants of Health: A Focus on Socioeconomic Position and Work Environment. Int J Environ Res Public Health. 2020;17(18). doi:10.3390/ijerph17186663
  23. Cortes-Ramirez J, Naish S, Sly PD, Jagals P. Mortality and morbidity in populations in the vicinity of coal mining: a systematic review. BMC Public Health. 2018;18(1):721. doi:10.1186/s12889-018-5505-7
  24. Naidoo R, Seixas N, Robins T. Estimation of respirable dust exposure among coal miners in South Africa. J Occup Environ Hyg. 2006;3(6):293-300. doi:10.1080/15459620600668973
  25. Oxman AD, Muir DC, Shannon HS, Stock SR, Hnizdo E, Lange HJ. Occupational dust exposure and chronic obstructive pulmonary disease. A systematic overview of the evidence. Am Rev Respir Dis. 1993;148(1):38-48. doi:10.1164/ajrccm/148.1.38
  26. Naidoo RN, Robins TG, Becklake M, Seixas N, Thompson ML. Cross-shift peak expiratory flow changes are unassociated with respirable coal dust exposure among South African coal miners. Am J Ind Med. 2007;50:992-998. doi:10.1002/ajim.20513
  27. Boyles AL, Blain RB, Rochester JR, et al. Systematic review of community health impacts of mountaintop removal mining. Environ Int. 2017;107:163-172. doi:10.1016/j.envint.2017.07.002
  28. Chapman AL, Hadfield M, Chapman CJ. Qualitative research in healthcare: an introduction to grounded theory using thematic analysis. J R Coll Physicians Edinb. 2015;45(3):201-205. doi:10.4997/JRCPE.2015.305
  29. Foley G, Timonen V. Using Grounded Theory Method to Capture and Analyze Health Care Experiences. Health Serv Res. 2015;50(4):1195-1210. doi:10.1111/1475-6773.12275
  30. Harris T. Grounded theory. Nurs Stand. 2015;29(35):32-39. doi:10.7748/ns.29.35.32.e9568
  31. Parsai MB, Castro FG, Marsiglia FF, Harthun ML, Valdez H. Using Community Based Participatory Research to Create a Culturally Grounded Intervention for Parents and Youth to Prevent Risky Behaviors. Prev Sci. 2011;12(1):34–47. doi:10.1007/s11121-010-0188-z